# Improving Recovery of Diatoms Bio-Silica Using Chemical Treatment with VAUS^TM^

**DOI:** 10.3390/ma17235742

**Published:** 2024-11-23

**Authors:** Se Ryung Suh, Joo Hun Lee, Gyung Min Go, Jaeyoung Lee, Hyunjoon Kong, Eun-Jin Park

**Affiliations:** 1Department of Prosthodontics, Graduate School of Clinical Dentistry, Ewha Womans University, Seoul 07985, Republic of Korea; durh9360@naver.com; 2Department of Chemical and Biomolecular Engineering, University of Illinois Urbana-Champaign, Urbana, IL 61801, USA; joohunl2@illinois.edu (J.H.L.); hjkong06@illinois.edu (H.K.); 3JDK Bio Inc., Jeju 14502, Republic of Korea; jdk8968@naver.com; 4Mirae Ultrasonic Tech., Bucheon 61801, Republic of Korea; forjylee@gmail.com; 5Department of Dentistry, College of Medicine, Ewha Womans University, Seoul 07804, Republic of Korea

**Keywords:** diatom, bio-silica, vacuum, ultrasonic, organic matter, *M. nummuloides*

## Abstract

High-temperature baking is a typical method to remove organic matter from diatoms, but it is not suitable for bio-silica because of the high crystallinity. This study provides a method using the VAUS^TM^ to remove organic matter from diatoms more quickly and biocompatibly. The optimal frequency for organic matter removal was investigated for domestically produced *M. nummuloides*. The removal efficiency of TOC/TN was calculated and analyzed. The C and Si elements were analyzed in EDS, and organic matter removal was analyzed by XRD. TOC RE% at a frequency of 35 kHz exhibited the highest value, indicating a statistically significant difference. XRD analysis demonstrated that the organic matter was nearly entirely removed using NaOCl compared to high-temperature-baked *M. nummuloides*. In the EDS analysis, there were significant differences in the C and Si elements with respect to frequency. This is very similar to the values from the positive control group, high-temperature-baked *M. nummuloides*. Ultrasonic treatment and frequency adjustments were found to significantly impact the chemical removal of organic matter from *M. nummuloides*. Although vacuum application was initially considered, it did not demonstrate a statistically significant effect according to TOC analysis.

## 1. Introduction

Recent societal trends show a rapid shift towards an aging population, with the elderly now constituting over 20% of the overall population. Accompanying this demographic transition is a noticeable surge in dental prosthetic users. Research involving Dental Health Care Professionals (DHCPs) from various nations and patients using dental prosthetics revealed that 1 in 7 DHCPs did not recommend primary methods due to a lack of evidence-based consensus. On the other hand, dental prosthetic users employed various cleansing methods, including saline, bleach, vinegar, and baking soda [1].

Biofilms are sessile microbial communities growing on surfaces, frequently embedded in a matrix of extracellular polymeric substances [2,3]. However, the surface of the acrylic resins can be colonized by a microbial biofilm, causing lesions of the buccal mucosa and favoring the development of systemic infections [4,5,6]. It is extremely important that chemical denture cleansers be used as an adjunct for patients unable to properly care for their dentures and manage overall oral hygiene [7]. Although Dental Cleaning Agents (DCAs) are effective in removing mild tartar or slight discoloration, mechanical cleansing methods remain essential and recommended due to the tenacious long-term adhesion of biofilm [8,9,10].

In this study, *Melosira nummuloides*, lava seawater diatoms primarily found during the low-temperature winter periods in the Northern Hemisphere’s temperate regions, were identified in certain areas of Jeju Island. These diatoms effectively block heavy metals and harmful substances, leading to reduced chances of contamination. Moreover, the lava seawater maintains consistent temperature and salinity levels throughout the year, negating the need for additional installations such as filtration, sterilization, or heating units for diatom processing. Notably, the procurement cost of lava seawater is 25–75 times less expensive compared to deep ocean water, making it a viable option for culturing purposes. Consequently, *M. nummuloides* was used as a negative control group in this study to compare organic matter removal rates under economically feasible mass culture conditions. Furthermore, this study aimed to explore the potential applications of constituents found in the mass-cultured *M. nummuloides*. By comparing properties based on different processing methods, it seeks ways to utilize them as high-value-added materials [11].

Bio-silica derived from *M. nummuloides* boasts a nano-porous structure, specifically a frustule, which provides an expansive surface area. Its resilience to strong acids and high temperatures renders it invaluable for diverse industries, including use as a photocatalyst, an adsorbing material, a template for nano-catalyst production, sensors, and a biocompatible material for biofilm removal.

However, a significant challenge arises in the initial step of producing industrially viable bio-silica, which involves the removal of organic matter [12,13]. This process often demands the excessive use of hazardous reagents, involves intricate procedures, and requires extended processing times. Moreover, during the extraction process, there is a notable risk of damaging the nano-porous structure. Bio-silica is currently extracted by treating *M. nummuloides* with 10% hydrochloric acid followed by high-temperature baking at 650 °C for 5 h. However, this method presents several challenges: it involves the use of high concentrations of acid, potentially increases cytotoxicity due to enhanced crystallinity from high-temperature baking, raises environmental pollution concerns, and necessitates an extended duration of the process. In dentistry, essential bone-graft materials used for procedures like implant installation face similar challenges. Conventionally, xenogenic bones such as bovine are treated with strong alkalis and subjected to heat treatment at temperatures from 300 °C to 1000 °C for at least 15 h, followed by gamma-ray sterilization. However, these methods can lead to issues with immune reactions and the osteogenic potential of the grafts. In response, techniques that utilize autogenous teeth as an alternative have been developed [14]. Inspired by this, our study aims to use periodic negative pressure and ultrasonication instead of high-temperature baking to remove organic matter from diatoms. The current limitation on experimental use in high-value applications, such as in medicine or optics, restricts its industrial application to less lucrative areas, like water purification filters. Our study introduces an eco-friendly method for the removal of organic matter from diatoms using NaOCl, which has the potential to revolutionize industry practices [15]. Our innovative approach combines the use of a vacuum ultrasonic device capable of periodically applying negative pressure to rapidly and biocompatibly produce bio-silica for diverse industrial applications. We aim to expedite the production process of bio-silica within this research’s scope and to mitigate the potential release of toxic substances associated with traditional combustion methods [16,17]. Notably, by evaluating organic matter removal rates at four distinct frequencies during the prototype phase, and adjusting vacuum and stirring durations, we will determine the optimal conditions for various applications. Such insights will undoubtedly enhance the material’s future applicability. The compact, low-noise design of our base unit allows for installation in confined spaces, facilitating its use in individual research endeavors. Furthermore, the reduced processing time presents a safe and efficient alternative for large-scale production, heralding a future where our breakthrough is widely adopted.

The aim of this study is to address the prevalent use of high-temperature baking for the removal of organic matter from diatoms. While this is a conventional method, it is not ideal for the formation of bio-silica due to its propensity to induce high crystallinity, compromising the material’s structural integrity. To overcome this limitation, we intend to introduce a more rapid and eco-friendly technique employing VAUS^TM^, a novel approach aimed at preserving structural integrity of bio-silica. This research seeks to pave the way for an innovative method that ensures optimal bio-silica production without compromising its structural integrity.

In dentistry, bio-silica can be utilized as an abrasive component in toothpastes or dental cleaning agents, owing to its ability to mechanically remove plaque and debris without causing damage to enamel. Furthermore, its nano-porous structure offers potential as a scaffold for drug delivery systems, enabling the controlled release of therapeutic agents in periodontal treatments. Beyond dental applications, bio-silica’s excellent biocompatibility and structural properties make it suitable for broader medical uses, such as bone tissue engineering and as a coating for medical implants to reduce microbial adhesion and biofilm formation. These diverse applications underscore the potential of diatom bio-silica as a versatile material in biomedical research and clinical practice.

## 2. Materials and Methods

The experimental raw-material, *Melosira nummuloides* (RM), which serves as the negative control group without any organic matter removal treatment, and the positive control group, bio-silica prepared by high-temperature baking, were provided by JDK Bio Co. (Jeju, Republic of Korea) (Figure 1). The *Melosira nummuloides* used in this study exhibits a frustule diameter ranging from approximately 9 to 42 μm, with a girdle band height between 10 and 14 μm. The frustule surface features a porous structure, with pore sizes ranging from 0.08 to 0.09 μm. These characteristics provide a high surface area relative to its weight, enabling efficient energy and material exchange. The average morphological dimensions were determined to ensure consistency in the samples used for analysis.

RM was produced from marine diatoms, *M. nummuloides*, isolated from lava seawater at concentrations of 2–4 cells/L. For large-scale cultivation, lava seawater was collected into a pond where *M. nummuloides* was cultured and harvested under optimal conditions. The harvested material was then washed with fresh water to desalinate and remove impurities, followed by natural dehydration to a moisture content of 85% or less through a 100 μm mesh filter. Bio-silica was produced from RM using methods based on freeze-dried *M. nummuloides* (FM) and ethanol-extracted *M. nummuloides* (EM). The process involved drying the diatoms at temperatures between 60 and 80 °C to remove moisture, followed by grinding. To eliminate organic matter, the material was treated with a hot solution of sulfuric acid/hydrogen peroxide or hydrochloric acid/hydrogen peroxide. Impurities were then removed by treating it with a 10% hydrochloric acid solution at 95 °C. After drying at 80 °C, the material was washed with distilled water at 65 °C, then vacuum dried at temperatures between 100 and 200 °C. The final product was diatom-derived bio-silica, with organic matter removed, ready for collection. The Vacuum Assisted Ultrasonic Stirrer (VAUS^TM^, Mirae Ultrasonic Tech., Bucheon, Republic of Korea) (Figure 2) utilized in this study can generate ultrasonic waves ranging from 35–170 kHz, in single or multiple frequencies. “+” indicates the presence of vacuum application, while “−” indicates the absence of vacuum application (Table 1).

A magnetic stirrer incorporated within the vacuum ultrasonic device enables the processing of lightweight particulate materials, potentially surpassing traditional equipment. This feature allows for both intensive and precision cleaning, ensuring a uniform distribution of energy across the entire sample and steady facilitation of reactions. The multiple-frequency ultrasonic generators, using specially distributed ultrasonic transducers within the device, deliver even energy throughout the sample, with each sample subjected to a single frequency at a time. It also reduces the cavitation size from the ultrasonic vibrator, which assures that the ultrasonic energy effectively penetrates the nano-porous structures, promoting simultaneous internal and external reactions. Negative pressure applied alongside the periodic creation of a vacuum by the vacuum pump swiftly removes simple bubbles, reducing cavitation blind spots and particle acceleration, thereby minimizing energy loss and enhancing the reaction process. The system also includes a magnetic stirrer that maintains fine particulate samples in suspension within the reaction solution, facilitating consistent and expedited cleaning and reactions.

In this study, 4.5% NaOCl was used to remove organic matter from the negative control group RM, instead of the conventional high-temperature baking or chemical solutions. After 50 g of RM was mixed with 500 mL of 4.5% NaOCl, the mixture was placed in the VAUS^TM^ device, which was stirred to prevent solidification. The settings of the VAUS^TM^ were adjusted based on frequency, vacuum application, and stirring time. Prolonged stirring increased the temperature within the VAUS^TM^ tank, necessitating the addition of ice water to maintain a temperature below 40 °C. The mixture was then divided into twelve 50 mL conical tubes, each containing 42 mL, and the contents were weighed to ensure even distribution. These tubes were centrifuged in a Combi 514R (Hanil, Seoul, Republic of Korea) at 4000 rpm for 15 min, and the supernatant (4.5% NaOCl) was discarded. The tubes were then paired, reducing the count to six and equalizing the volume to 45 mL with the addtion of DI water. This was followed by a 10-min vacuum ultrasonic cleansing session (Flexonic, Mirae Ultrasonic Tech., Republic of Korea). Subsequent to cleansing, the six conical tubes were centrifuged again under the same conditions, and the supernatant (DI water) was removed. The remaining tubes were combined into pairs to form three tubes, equalized to 45 mL with DI water, and manually shaken for further cleansing. After a final centrifugation at 4000 rpm for 15 min, the supernatant (DI water) was discarded, and the tubes were stored at a temperature of 10 to 15 °C.

The centrifuged samples were rapidly frozen using a freeze dryer (Bondiro, Ilsin lab, Yangju, Republic of Korea) and, with the assistance of a vacuum pump, moisture and other components were removed over 48 h at temperature typically lower than 10 °C, achieving freeze-drying. Equipped with a high-capacity sealed freezing system, the chamber’s interior was quickly cooled to below −40 °C. The samples were dried using a method that involves rolling the flasks, which completely eliminated any chances of effervescence and melting. This method promoted a film-like preliminary freeze, suppressing the surface hardening and concentrated solidification of the samples during freezing, resulting in an optimal drying outcome. After freeze-drying, the samples were sealed with parafilm to prevent moisture and foreign substances from entering. Subsequent analyses and measurements for TOC/TN, EDS, and XRD data were conducted on these samples.

SEM images were captured using Field-Emission Scanning Electronic Microscopy (AURIGA, Carl Zeiss, Jena, Germany) to observe the porous structure, particle size, and microstructure of *M. nummuloides* and the sample after organic matter removal.
$$TOC\;RE\%=\frac{TOCi-TOCf}{TOCi}\times 100$$
$$TN\;RE\%=\frac{TNi-TNf}{TNi}\times 100$$

All samples, including the negative and positive control groups, were analyzed for Total Organic Carbon (TOC) and Total Nitrogen (TN) using the TOC Analyzer (Sievers 5310 C, GE Analytical Instruments, Boulder, CO, USA). In the TOC analysis, samples were processed in the UV oxidation chamber, where organic matter was fully oxidized to carbon dioxide (CO_2_). The CO_2_ produced was then captured through a selective membrane, and its quantity was measured using a conductivity detector. In the TN analysis, inorganic carbon dissolved in the sample was completely converted to CO_2_ under acidic conditions. Only the CO_2_ that passed through the selective membrane was captured, and the total nitrogen amount was measured using the conductivity detector. The microprocessor determined the exact amount of TOC by calculating the difference between the total carbon and the inorganic carbon. The analyzed TOC and TN values are denoted as TOCi and TOCf for initial (negative control group) and final (sample) TOC, and TNi and TNf for initial (negative control group) and final (sample) TN values, respectively. These values were then used in an equation to compute the TOC and TN removal efficiency (RE%).

XRD analysis was performed using Powder X-Ray Diffractometry (D8 Advance, Bruker, Karlsruhe, Germany). Cooper K-alpha radiation was directed at the sample, and the resulting diffraction patterns facilitated the structural analysis of crystalline and amorphous materials, including phase analysis and crystal orientation. In this study, XRD was employed to elucidate the characteristics and structural attributes of frustules before and after the NaOCl process. To evaluate the efficiency of organic matter removal, analyses of the crystalline structures of the untreated frustule (UF) and the treated frustule (TF) were conducted.

EDS analyses were conducted using Field-Emission Scanning Electronic Microscopy (AURIGA, Carl Zeiss, Germany). Measurements were based on the characteristics of secondary electrons and backscattered electrons released after the electron beam interacted with the sample, forming a three-dimensional image. The weight percent (wt.%) of the carbon (C) element was compared to gauge the amount of organic matter before and after ultrasonic treatment. Following NaOCl treatment, the organic matter was removed. Through EDS analysis, it was possible to compare the weight and ratio changes of the C and Si elements during the NaOCl treatment.

All statistical analyses were performed using SPSS for Window (SPSS version 29; IBM Corporation, Armonk, NY, USA). The data’s distribution was evaluated for normality using Kolmogorov–Smirnov and Shapiro–Wilk tests. To compare the means of each experimental group and the control group in the TOC/TN and EDS analyses, *t*-tests and one-way ANOVA followed by Scheffe’s post hoc tests were performed. *p*-values less than 0.05 were considered to reflect statistically significant differences.

## 3. Results

### 3.1. SEM Images of M. nummuloides

SEM images of *M. nummuloides* were captured after the removal of organic matter at different frequencies. Overall, they presented a tendency for some parts to detach, yet no distinct differences were observed between the groups. The average diameter of these particles was measured to be 100–500 nm (Figure 3). However, as the SEM images alone did not provide sufficient visual information to compare the efficiency of organic matter removal between the experimental groups, additional Total Organic Carbon (TOC) and Total Nitrogen (TN) analyses were conducted to quantify the differences in organic content.

### 3.2. Total Organic Carbon (TOC) and Total Nitrogen (TN) Analysis

Total Organic Carbon (TOC) analysis did not reveal significant differences based on vacuum application or stirring time. However, at 35 kHz, the TOC removal efficiency was the highest at 83.4 ± 3.0, with a *p*-value of 0.008, indicating a statistically significant difference. This suggests that ultrasonic treatment at 35 kHz is particularly effective in enhancing organic matter removal. In contrast, for Total Nitrogen (TN), no significant differences were observed with respect to vacuum application, stirring time, or frequency, as all *p*-values were above 0.05 (Table 2). This result indicates that factors such as ultrasonic frequency and vacuum application had a lesser impact on nitrogen removal compared to their effects on TOC.

### 3.3. XRD Analysis

XRD analysis showed that organic matter was almost entirely removed using 4.5% NaOCl compared to the high-temperature-baked *M. nummuloide* (Figure 4).

XRD analysis was performed to assess the efficiency of organic matter removal across the experimental and control groups. The results revealed distinct differences in the characteristic peaks between the groups, specifically in the regions associated with carbon ((CH)N), silicon, and oxygen. In the negative control group, prominent peaks were observed in the (CH)N region, indicating the presence of residual organic matter. These peaks were accompanied by weak SiO_2_-related signals, suggesting minimal contribution from inorganic components. The positive control group, which underwent treatment, showed a noticeable reduction in the (CH)N peaks and a slight increase in the intensity of SiO_2_ peaks, reflecting partial removal of organic matter. In contrast, the experimental group, treated under optimized conditions (35 kHz, 60 min, vacuum+), exhibited a near-complete absence of (CH)N peaks. This result indicates the effective removal of organic matter. Furthermore, the SiO_2_ peaks were significantly more pronounced compared to both control groups, confirming that the treatment enhanced the presence of inorganic compounds. Notably, the Fe3Si peak remained consistent across all groups, indicating that the stability of inorganic phases was unaffected by the treatment process.

### 3.4. EDS Analysis

In the EDS analysis, the weight percent (wt.%) of carbon (C), silicon (Si), and oxygen (O) elements showed no statistically significant differences based on vacuum application or stirring time. However, at 35 kHz, significant differences were observed: the weight percent for the C element was 45.9 ± 8.6, and, for the Si element, it was 20.5 ± 5.2. The *p*-values were 0.024 for C and 0.012 for Si, both below 0.05, indicating statistically significant differences. These findings are in close agreement with those of the positive control group, high-temperature-baked *M. nummuloides*, which showed a C weight percent of 44.6 and a Si element percent of 20.1 (Table 3).

#### 3.4.1. SEM and EDS Element Mapping Analysis

The SEM images and EDS element mapping (A, B, C) reveal distinct morphological and elemental differences among the groups. Negative Control (A): The SEM image of the raw *M. nummuloides* (A) shows a smooth surface morphology, consistent with the presence of intact organic matter. The EDS mapping shows prominent carbon (C) signals and relatively low silicon (Si) intensity, confirming a substantial amount of organic material and limited exposure of the silica structure. Positive Control (B): The high-temperature-baked *M. nummuloides* (B) exhibits a more porous structure, indicating partial degradation of the organic layer. The EDS mapping displays reduced carbon and increased silicon signals, suggesting effective but incomplete organic matter removal during baking. Experimental Group (C): The experimental group treated with 35 kHz, 60 min, and vacuum+ conditions shows a highly porous silica structure with minimal organic residue. The EDS mapping demonstrates a significant reduction in carbon signals and enhanced silicon and oxygen intensities, confirming near-complete removal of organic matter.

#### 3.4.2. EDS Spectra Analysis

The EDS spectra (A, B, C) further validate the differences observed in the SEM and mapping analysis. Negative Control (A): high carbon peaks (~45%) dominate the spectrum, with lower silicon and oxygen peaks (~18% and ~12%, respectively). Positive Control (B): carbon peaks are reduced (~30%), while silicon and oxygen peaks increase to ~25% and ~18%, respectively, indicating partial organic removal. Experimental Group (C): carbon content decreases significantly (~15%), while silicon and oxygen peaks are the most prominent (~35% and ~30%, respectively). This highlights the superior efficacy of the experimental treatment in exposing the silica structure. (Figure 5). This suggests that the NaOCl treatment combined with 35 kHz ultrasonic frequency is highly effective in removing organic matter, achieving results comparable to high-temperature baking.

## 4. Discussion

There are various methods for maintaining hygiene and disinfecting materials, including mechanical disinfection (e.g., scrubbing with toothpaste or soap), chemical disinfection (e.g., soaking in disinfectant solutions), and physicochemical methods (e.g., UV radiation and ultrasound), as well as combinations of these techniques [18]. In previous studies, ultrasonic cleaning, particularly in combination with surfactants, has been shown to be effective in disrupting biofilms and bacterial clusters, which are resistant to standard chemical treatments [19]. In dental applications, ultrasonic irrigation of dentures has significantly reduced bacterial colonization and decreased the incidence of aspiration pneumonia among elderly individuals dependent on such care [20]. Similarly, in our study, we aimed to leverage the advantages of ultrasonic cavitation to enhance the cleaning efficiency of *M. nummuloides* diatoms, which are known for their porous frustule structure.

Mechanism of Organic Matter Removal: The combination of NaOCl and ultrasonic frequency (35 kHz) effectively removes organic matter through cavitation and oxidation. Ultrasonic cavitation generates microbubbles that collapse near the diatom surface, producing localized high-energy impacts that disrupt organic layers [21]. This mechanical effect is complemented by the oxidative power of NaOCl, which degrades residual organic compounds chemically, as confirmed by our XRD and EDS analyses. This combined approach allows for efficient organic matter removal without damaging the diatom’s frustule structure, thereby preserving its integrity for potential applications in catalysis and adsorption.

Comparison with Other Organic Matter Removal Methods: Various cleaning agents, such as sulfuric acid [22,23], hydrogen peroxide [24,25,26], and nitric acid [27], have been used for oxidant cleansing. However, in our pre-tests, 4.5% NaOCl combined with ultrasonic cavitation outperformed these conventional oxidants. For instance, while treatments with peracetic acid (PAA) and FeSO_4_-based Fenton processes showed incomplete removal of organic matter, the NaOCl–ultrasonic combination achieved up to 97% TOC removal and 99% TN removal, demonstrating superior efficacy. This suggests that NaOCl’s strong oxidizing capability, combined with ultrasonic cavitation’s physical disruption, provides a more effective cleaning solution for diatom frustules.

Advantages Over High-Temperature Baking: Traditional high-temperature baking (400–800 °C) is widely used to remove organic matter from diatom frustules but often results in increased crystallinity, which can hinder the functional properties of bio-silica [28]. High-temperature baking not only consumes significant energy but may also produce toxic by-products [29]. In contrast, our VAUS^TM^ method maintains an amorphous silica structure, which is preferable for applications requiring high surface area and porosity. Moreover, this approach minimizes environmental impact by avoiding extreme temperatures and reducing resource consumption.

Despite numerous attempts under various conditions, no experimental variable demonstrated markedly distinct results. However, a general trend was observed: carbon was effectively removed at low frequencies, and the results support this observation. For nitrogen, although removal appeared more efficient at high frequencies, the results are inconclusive due to a high *p*-value and fluctuations in TN RE% with increasing frequency. This indicates that the physical force from cavitation plays a significant role in removing organic matter. In contrast, nitrogen appears to be eliminated by dissolving into the solution, particularly in the form of NaOCl. In such scenarios, it is the activation of the solution by agitation or vibration, rather than the intensity of cavitation, that becomes critical, and, while higher frequencies may suggest a potential advantage, the data do not conclusively support this observation. Notably, studies in wastewater treatment often use ultrasonic ranges of 300–500 kHz. The organic matter removal in our experiment, while slightly inferior to that of the high-temperature-baking positive control group, was still comparable. One possible explanation is that high-temperature baking increases diatom crystallinity, which prevents organic matter from re-adhering to the diatom surface. However, mechanical and chemical treatments using VAUS^TM^ maintain an amorphous state, suggesting that the removed organic matter might reattach due to electrostatic forces. To address these limitations, it is postulated that employing plasma treatment could enhance organic matter removal rates. In terms of nitrogen removal, many samples exhibited near-complete removal rates of up to 97.21%, slightly exceeding the 98.7% observed in the positive control group treated with high-temperature baking. The removal efficiency was generally higher at lower frequencies, and longer stirrer times tended to improve TOC removal efficiency. Although vacuum application was also tested, it did not show a statistically significant effect due to high *p*-values.

Statistically, lower frequencies significantly correlated with greater removal efficiency. Among various organic matter removal solutions, including ferrous iron (Fe(II)), peracetic acid (PAA), the Fenton process, and sono-Fenton, the application of 4.5% NaOCl proved most effective. Based on pre-test comparisons of these five solutions, 4.5% NaOCl demonstrated the highest efficiency in organic matter removal, making it the most suitable option among the treatments tested. Specifically: Fenton-type (PAA) treatment: organic matter removal was incomplete, as the treated sample did not exhibit the characteristic white color of bio-silica, indicating that PAA was insufficient for full organic matter removal. Sono-Fenton (FeSO_4_) treatment: even after 1–3 h of treatment at 35 kHz, TOC and TN removal efficiencies did not exceed 85%, likely due to the diatoms’ size and morphological differences impacting the organic matter decomposition state. 1.3% NaOCl treatment: more effective than the sono-Fenton process, this treatment resulted in higher levels of organic matter removal but was still not as effective as higher concentrations of NaOCl. Sono-1.3% NaOCl treatment: TOC removal efficiency reached 95%, and TN removal efficiency reached 98% after 2 h of treatment, but no further improvement was observed with extended treatment times. Sono-4.5% NaOCl treatment: this solution achieved the highest efficiency, with 97% TOC removal after 2 h and 99% TN removal after 1 h. While the periodic vacuum technique did not enhance the results, the 35 kHz ultrasonic frequency showed a statistically significant effect on removal efficiency. These pre-test results highlight 4.5% NaOCl under ultrasonic conditions as the most effective treatment option, demonstrating superior removal of organic matter compared to the other solutions tested.

Coupling 4.5% NaOCl with mechanical stirring, ultrasonic application, and periodic negative pressure seemed to enhance organic matter removal from even the fine powder particles and the frustule pore structures.

The XRD analysis results demonstrate the efficacy of the experimental treatment in removing organic matter. The prominent (CH)N peaks in the negative control group highlight the presence of residual organic compounds in untreated samples. In contrast, the marked reduction of these peaks in the experimental group provides strong evidence that the treatment successfully removed organic content. The increased intensity of SiO_2_-related peaks in the experimental group further supports this finding. SiO_2_ is known to remain as a stable inorganic compound following the removal of organic material, and its prominence in the experimental group indicates that the treatment enhanced the exposure of silica after organic content was eliminated. This aligns with this study’s objective of confirming organic matter removal through elemental analysis of carbon, silicon, and oxygen. The Fe3Si peak, which remained consistent across all groups, confirms that the inorganic structural integrity was preserved during the treatment. This is an important finding, as it demonstrates that the removal of organic matter did not disrupt the underlying inorganic matrix, a key requirement for applications where structural stability is critical. Overall, these findings validate the effectiveness of the experimental conditions (35 kHz, 60 min, vacuum+) in achieving substantial organic matter removal while preserving the stability of inorganic components. This treatment approach may offer a reliable solution for processes requiring efficient organic matter elimination without compromising inorganic material integrity.

The combination of SEM imaging, EDS mapping, and EDS spectra provides strong evidence for the effective removal of organic matter in the experimental group compared to the control and positive control groups. In negative control group (A), the smooth morphology and high carbon content in both the SEM and EDS data confirm the untreated state of raw *M. nummuloides*. The low silicon intensity further supports the presence of a thick organic layer that obscures the silica structure. In positive control group (B), the high-temperature-baked sample shows partial organic matter removal, as indicated by reduced carbon peaks and increased silicon exposure. However, the results suggest that the baking process may not fully eliminate organic matter or preserve the silica structure in optimal condition. In experimental group (C), the treatment at 35 kHz, 60 min, and vacuum+ conditions shows the most substantial removal of organic matter, as evidenced by minimal carbon signals and prominent silicon and oxygen intensities in the EDS spectra. The SEM images corroborate this, displaying a well-defined and porous silica structure. These findings demonstrate that the ultrasonic treatment achieves comparable or superior results to high-temperature baking while preserving the silica framework more effectively. The non-thermal approach (ultrasonic and NaOCl treatment) offers significant advantages over high-temperature baking. It effectively removes organic content without risking structural compromise, making it a promising method for applications requiring silica integrity.

The use of sodium hypochlorite (NaOCl), commonly employed for dental root canal irrigation, resulted in the highest organic matter removal efficiency when compared to the Fenton-type process using peracetic acid (PAA) and the traditional Fenton process with ferrous iron and hydrogen peroxide. The removal efficiency was found to vary with the frequency and material used, with longer stirring times and the application of a vacuum being correlated with improved removal efficiencies. The VAUS^TM^ developed in this study is anticipated to offer an environmentally friendly solution for producing high-quality amorphous bio-silica. It removes organic matter more quickly than traditional combustion or high-temperature baking methods. This system has enabled mechanical cleansing, which was previously impossible with conventional dental cleansing agents, marking a significant advancement—especially for the elderly with mobility issues, potentially enhancing their overall oral health.

While this study primarily relied on XRD, EDS, and SEM for the characterization of organic matter removal and silica exposure, we recognize that additional characterization techniques, such as Fourier Transform Infrared Spectroscopy (FTIR), could further enhance our understanding. FTIR, for example, would allow for the identification of functional groups associated with organic compounds and provide a molecular-level confirmation of organic matter removal. Although FTIR analysis was not included in the current study due to resource and time constraints, we plan to incorporate this technique in future research. Such additional characterization methods will provide complementary insights into the efficiency of the treatment process and support the development of a more detailed understanding of the structural and chemical changes in the material.

Given the current shortage of effective chemical cleansing methods for biofilm eradication, the clinical application of VAUS^TM^ could lead to transformative outcomes for oral and systemic health. Additionally, VAUS^TM^ not only shortens the time needed to produce bio-silica but also significantly reduces the potential release of various toxic substances, addressing a major concern associated with traditional high-temperature baking methods. The integration of a magnetic stirrer in this innovative device indicates its broad applicability for processing smaller, lightweight particle materials in various ways compared to current equipment. With extensive research into four different frequencies at the prototype stage, determining the optimal conditions for specific applications will undoubtedly increase its utility in the future. The device’s compact and low-noise design makes it suitable for use in confined spaces and for individual research. Moreover, the reduced processing time suggests that VAUS^TM^ can be a safe and rapid alternative for mass production when needed.

## 5. Conclusions

Based on the experimental conditions of our study, the application of ultrasonic treatment and the adjustment of its frequency have proven to be effective methods for chemically removing organic matter from *M. nummuloides*. Although vacuum application was tested during NaOCl treatment, TOC analysis did not reveal significant differences based on vacuum application alone. VAUS^TM^ could be a valuable device for effectively removing organic matter from diatoms. With periodic vacuum ultrasonication and the aid of a magnetic stirrer, this experiment demonstrated the capability to remove organic matter comparably to the existing high-temperature baking process but in a shorter time and using environmentally friendly chemical agents. This method holds promise for synthesizing high-quality amorphous bio-silica suitable for various applications, addressing the limitations of chemical treatments that use high-concentration toxic solutions—known for their inferior ability to remove organic matter—and heat treatments that can increase crystallinity.

## Figures and Tables

**Figure 1 materials-17-05742-f001:**
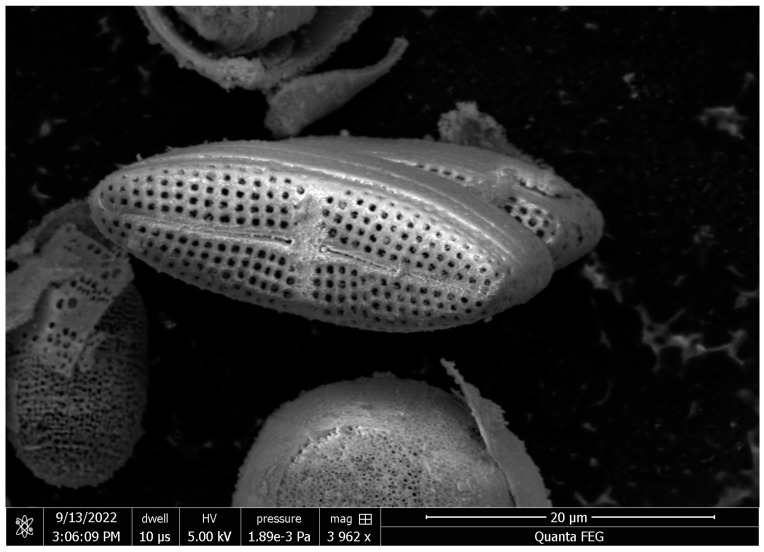
*Melosira nummuloides* from the sea of Korea (JDK Bio Co., Jeju, Republic of Korea).

**Figure 2 materials-17-05742-f002:**
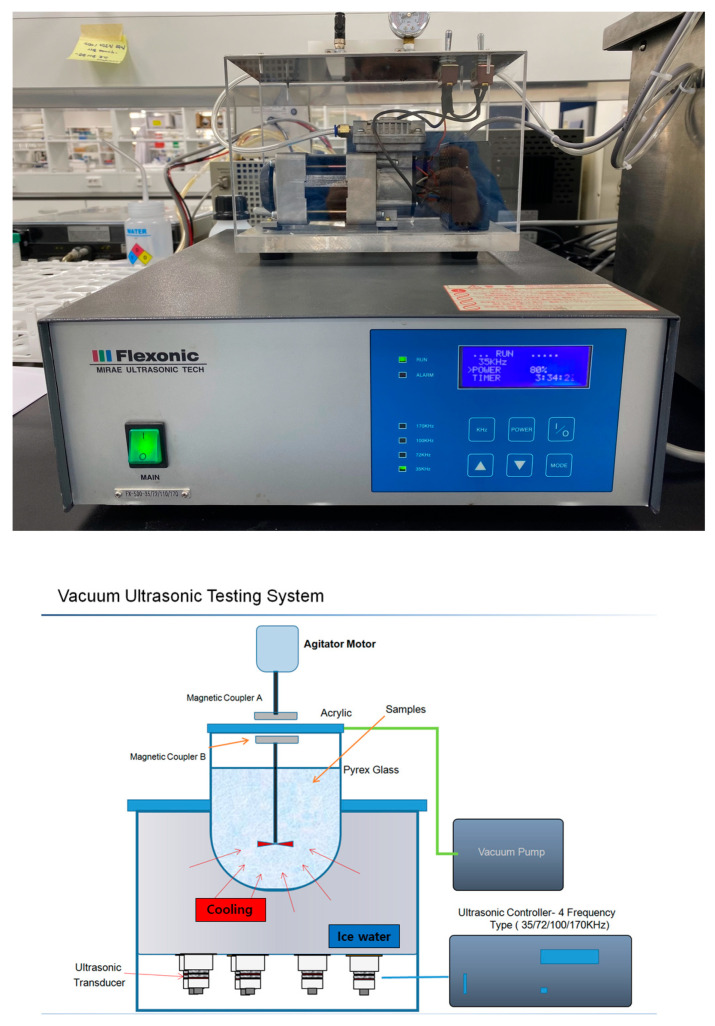
Vacuum Assisted Ultrasonic Stirrer (VAUS^TM^, Mirae Ultrasonic Tech., Republic of Korea).

**Figure 3 materials-17-05742-f003:**
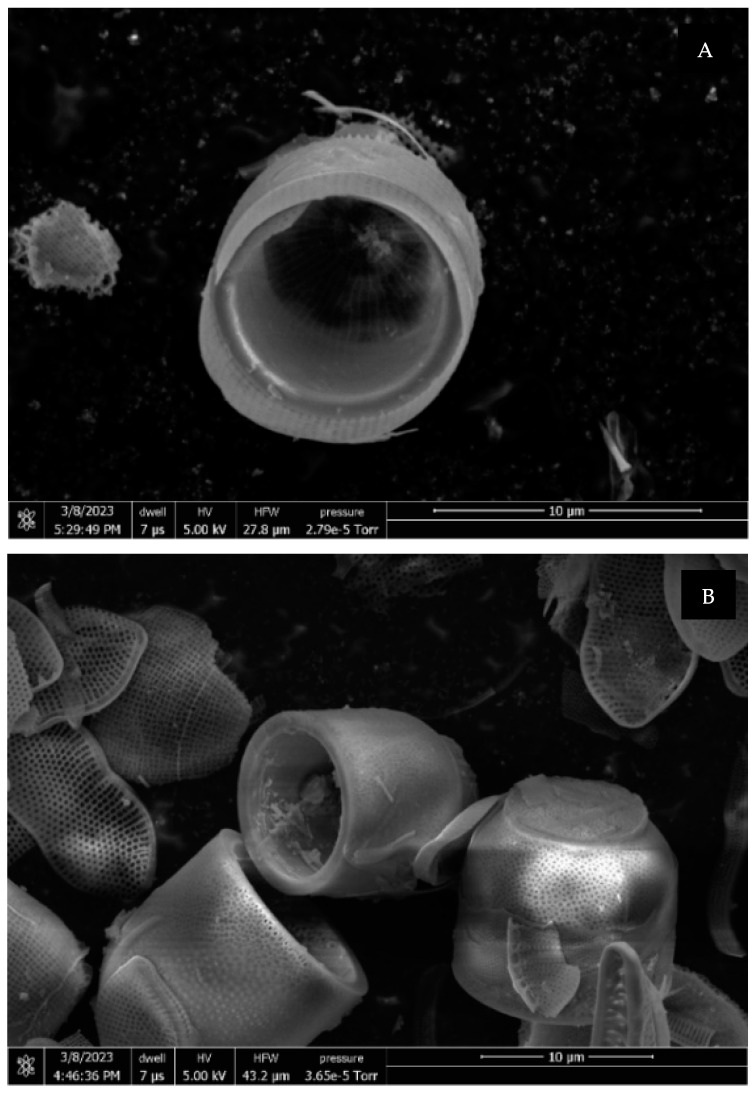
SEM images of *M. nummuloides* after organic matter removal at different frequencies. (**A**) 35 kHz; (**B**) 72 kHz; (**C**) 100 kHz; (**D**) 170 kHz.

**Figure 4 materials-17-05742-f004:**
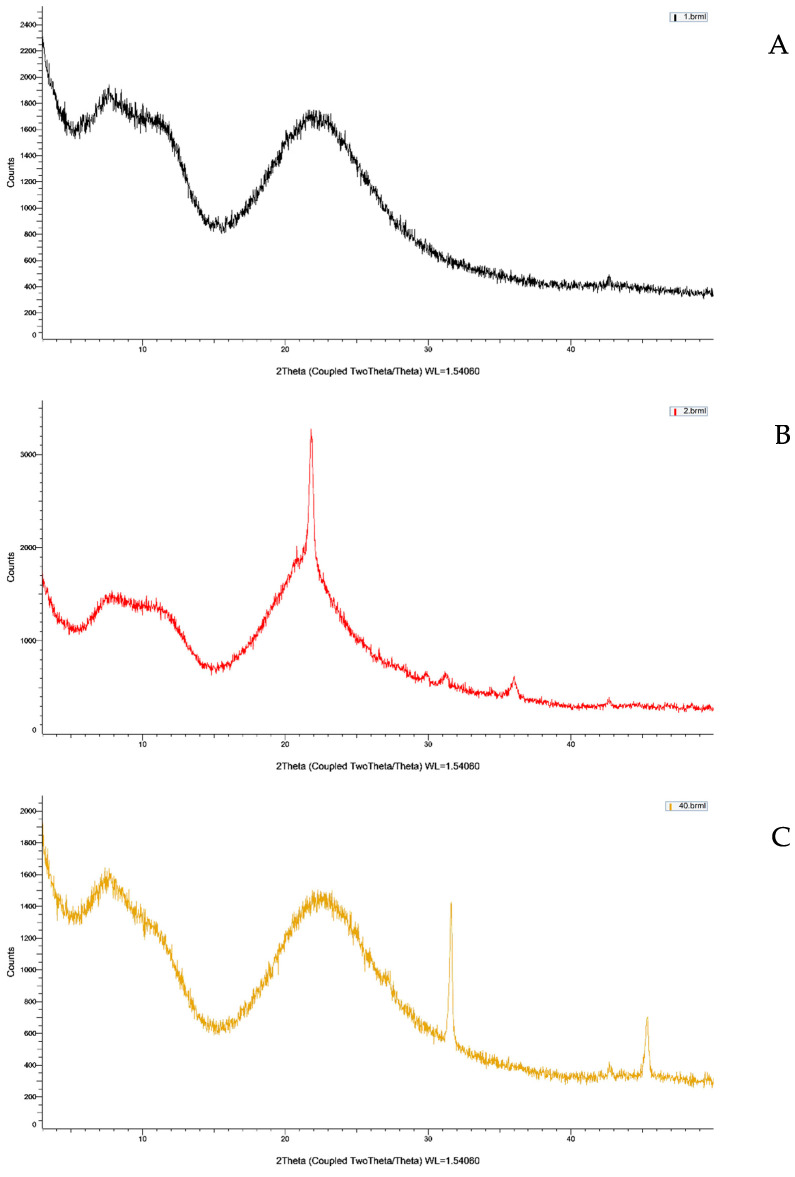
XRD images of frustule crystal structure analysis of *M. nummuloides* after organic matter removal at different time, frequencies, and vacuum application. (**A**) Negative control: RM (Raw-material *Melosira*). (**B**) Positive control: high-temperature-baked *Melosira*. (**C**) 35 kHz, 60 min, vacuum+ *Melosira* after organic matter removal. Peak identification and indexing were conducted using the ICDD PDF-4+ database. The crystalline SiO_2_ phase was identified using 01-070-3755 (alpha-quartz) and (CH)N-related phase; graphitic carbon was referenced using PDF 01-075-2078 to identify residual organic content.

**Figure 5 materials-17-05742-f005:**
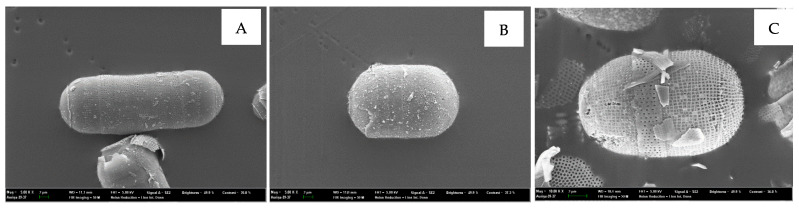
SEM images and EDS element mapping of *M. nummuloides* after organic matter removal. (**A**) Negative control: RM (raw-material *Melosira*). (**B**) Positive control: high-temperature-baked *Melosira*. (**C**) 35 kHz, 60 min, vacuum+ *Melosira* after organic matter removal.

**Table 1 materials-17-05742-t001:** Classification of groups for organic matter removal efficiency evaluation.

Groups	Different Condition		
**Negative control**	RM (Raw-material *Melosira*)		
**Positive control**	High-temperature-baked *Melosira* (65 °C, 5 h)		
**Experimental group**	Ultrasonic Frequency (kHz)	Vacuum (+/−)	Stirrer Time (min)
	35 kHz	+	15 min
	72 kHz		30 min
	100 kHz	−	
	170 kHz		60 min

**Table 2 materials-17-05742-t002:** Measured values of Total Organic Carbon (TOC) and Total Nitrogen (TN) RE (%) analysis.

**TOC RE (%)**	**Experimental Group**		**Mean ± SD**	***p* Value**
	Vacuum application (+/−)	+	81.5 ± 3.2	0.599 *
		−	82.0 ± 3.9	
	Sterring time (min)	15	80.9 ± 3.4	0.183 **
		30	81.3 ± 3.8	
		60	83.1 ± 3.2	
	Frequency (kHz)	35	83.4 ± 3.0	0.008 **
		72	82.8 ± 2.1	
		100	81.9 ± 3.2	
		170	78.9 ± 4.2	
**T** **N RE (%)**	**Experimental group**		**Mean ± SD**	***p* value**
	Vacuum application (+/−)	+	97.9 ± 4.2	0.904 *
		−	97.8 ± 4.1	
	Sterring time (min)	15	98.3 ± 1.9	0.183 **
		30	97.2 ± 4.9	
		60	98.1 ± 4.9	
	Frequency (kHz)	35	97.7 ± 5.9	0.757 **
		72	98.1 ± 3.0	
		100	96.9 ± 5.0	
		170	98.7 ± 1.4	

* T-test; ** one-way ANOVA; positive control TOC RE: 100%. positive control TN RE: 97.21%.

**Table 3 materials-17-05742-t003:** Measured values of C, Si, O mass norm (%) in EDS analysis.

**C** **Mass norm. (%)**	**Experimental Group**		**Mean ± SD**	***p* Value**
	Vacuum application (+/−)	+	48.9 ± 9.5	0.313 *
		−	51.4 ± 6.5	
	Sterring time (min)	15	51.1 ± 9.1	0.699 **
		30	47.7 ± 8.0	
		60	51.7 ± 7.2	
	Frequency (kHz)	35	45.9 ± 8.6	0.024 **
		72	47.5 ± 8.1	
		100	52.8 ± 6.7	
		170	54.4 ± 6.7	
**Si** **Mass norm. (%)**	**Experimental group**		**Mean ± SD**	***p* value**
	Vacuum application (+/−)	+	16.9 ± 6.4	0.733 *
		−	16.4 ± 4.0	
	Sterring time (min)	15	16.9 ± 5.7	0.729 **
		30	17.3 ± 5.5	
		60	15.8 ± 4.7	
	Frequency (kHz)	35	20.5 ± 5.2	0.012 **
		72	17.0 ± 5.8	
		100	15.2 ± 3.8	
		170	14.0 ± 4.2	
**O** **Mass norm. (%)**	**Experimental group**		**Mean** **± SD**	***p* value**
	Vacuum application (+/−)	+	34.1 ± 6.9	0.200 *
		−	32.1 ± 2.7	
	Sterrring time (min)	15	31.9 ± 3.5	0.331 **
		30	35.0 ± 7.8	
		60	32.4 ± 2.7	
	Frequency (kHz)	35	33.5 ± 3.5	0.214 **
		72	35.5 ± 8.9	
		100	31.8 ± 3.0	
		170	31.5 ± 2.7	

* T-test; ** one-way ANOVA; positive control C mass norm: 65.50%; positive control Si mass norm: 7.56%; positive control O mass norm: 26.94%.

## Data Availability

The datasets used and/or analyzed during the current study are available from the corresponding author on reasonable request.

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
