# Peer review of "Improving Recovery of Diatoms Bio-Silica Using Chemical Treatment with VAUSTM"

_materials, 2024, doi:10.3390/ma17235742_

Round 1
Reviewer 1 Report (Previous Reviewer 1)
Comments and Suggestions for Authors
Revier comments (2nd Revision) on Materials paper by Se Ryung Suh et al - Improving Recovery of Diatoms Bio-Silica…:
1) Figure-4: The labels to identify the different color curs are missing. How are they related to different time, frequencies and vacuum application? – may be explained in detailed intext. Nothing is clear at the (CH)N and Fe3Si peaks as so many color curves overlap in narrow region. I suggest reducing number of curves to 2 to 3 by selecting the one remarkable to support the point in discussion.
2) Figure 5 is confusing. First, it is not an elemental mapping but elemental analysis (elimental mapping is when you combine EDX and SEM results to show surface images with elemental distribution, ref.: https://semfe.stanford.edu/applications/EDS_Map). Identifying C amount variations from 5 is unclear as it is close to (overlap with) Pt peak. All C, O, and Si peaks are higher in 5D compared to 5A. All these peaks are lower in 5C compared to 5A. So, the statement “EDS analysis, which assesses elemental distribution on the sample surface, showed a decrease in carbon and an increase in silicon relative to the original melosira” in discussions is incorrect. If the Figure 5 peaks are not to scale comparable, one needs to analyze the spectra to give quantitative values, there is no need to show the confusing Figure 5, instead provide the quantitative results to support the statement.
I suggest removing Figure 5, including the quantitative data in the discussion, and restore back the Figure-5 SEM images from the previous version of the paper (this is more meaningful).
3) Page-16, 3rd paragraph: “In terms of nitrogen removal, many samples exhibited a complete removal rate of up to 100%, outforming the positive control group treated with high-temperature baking”. 100% is not correct, it is only slightly higher 97.21% vs 98.7% maximum.
4) Better to give the values for Positive control in Table 3 as well as given in Table 2 (at the bottom of the table).
5) Page-16, 4th paragraph: the first line “Statistically, higher frequencies significantly correlated with greater removal efficiency” is incorrect and contradicts with the last line in the above paragraph “The removal efficiency was generally higher at lower frequencies……”.
Author Response
Comments 1 : Figure 4: The labels to identify the different color curs are missing. How are they related to different time, frequencies and vacuum application? – may be explained in detailed intext. Nothing is clear at the (CH)N and Fe3Si peaks as so many color curves overlap in narrow region. I suggest reducing number of curves to 2 to 3 by selecting the one remarkable to support the point in discussion.
Response1 : Thank you for your insightful feedback. Based on your suggestions, we have revised the Resultsand Discussionsections to provide greater clarity and align them more closely with the presented data. Specifically, we have incorporated a detailed explanation of the three key graphs (negative control, positive control, and experimental group) and highlighted the changes observed in the (CH)N and Fe3Si peak regions.
Results Section: We revised the description to explicitly compare the (CH)N peaks across the negative control, positive control, and experimental group. The negative control showed strong (CH)N peaks, while the positive control showed reduced intensity, and the experimental group demonstrated a near-complete absence of these peaks, indicating effective organic matter removal.
The SiOâ‚‚ peaks were analyzed in more detail, emphasizing their increased prominence in the experimental group as evidence of organic matter removal. The Fe3Si peaks were noted to remain consistent across all groups, reflecting the stability of the inorganic structure.
Discussion Section: The revised discussion now contextualizes the results, linking the reduction in (CH)N peaks and the enhancement of SiOâ‚‚ peaks to the successful removal of organic matter in the experimental group (35 kHz, 60 min, vacuum+).
We expanded the interpretation of the Fe3Si peak consistency, emphasizing that the treatment preserved the stability of inorganic components, which is critical for the integrity of the overall material. These findings were connected back to the study's objectives, showcasing the significance of the observed changes and validating the effectiveness of the experimental treatment.
We believe these revisions improve the overall presentation and interpretation of the data. The updated sections now provide a clearer explanation of the observed results and their implications, as reflected in the three key graphs. Thank you again for your valuable input, which helped us enhance the quality of our manuscript. The changes can be found on page 8-9, lines 335-356 and page 13-14, lines 505-542 in the revised manuscript.
Comments 2: Figure 5 is confusing. First, it is not an elemental mapping but elemental analysis (elimental mapping is when you combine EDX and SEM results to show surface images with elemental distribution, ref.: https://semfe.stanford.edu/applications/EDS_Map). Identifying C amount variations from 5 is unclear as it is close to (overlap with) Pt peak. All C, O, and Si peaks are higher in 5D compared to 5A. All these peaks are lower in 5C compared to 5A. So, the statement “EDS analysis, which assesses elemental distribution on the sample surface, showed a decrease in carbon and an increase in silicon relative to the original melosira” in discussions is incorrect. If the Figure 5 peaks are not to scale comparable, one needs to analyze the spectra to give quantitative values, there is no need to show the confusing Figure 5, instead provide the quantitative results to support the statement.
I suggest removing Figure 5, including the quantitative data in the discussion, and restore back the Figure-5 SEM images from the previous version of the paper (this is more meaningful).
Response 2 : Thank you for your insightful comments and suggestions. Based on your feedback, we have revised our experimental analysis and incorporated SEM imaging and EDS analysis for clearer comparison between the different sample groups. Specifically, we focused on three key groups:
Negative Control (Raw M. nummuloides): This group represents untreated samples, serving as a baseline for assessing organic matter presence and structural integrity.
Positive Control (High-temperature baked M. nummuloides): This group provides a reference for the effects of conventional high-temperature treatment on organic matter removal.
Experimental Group (35 kHz, 60 min, vacuum+ treated M. nummuloides): This group demonstrates the effectiveness of the proposed ultrasonic treatment for organic matter removal.
Updates Made:
SEM Imaging: SEM images were captured for all three groups to visualize the morphological differences. The negative control shows an intact and smooth organic layer. The positive control exhibits partial removal of the organic layer, resulting in a porous structure. The experimental group displays a highly porous and well-defined silica structure, indicating significant organic matter removal with minimal residue.
EDS Analysis: Elemental analysis was performed on the same groups to quantify the changes in carbon, silicon, and oxygen content.
Negative Control: High carbon content with low silicon and oxygen signals, confirming the presence of organic matter.
Positive Control: Reduced carbon content and increased silicon and oxygen signals, indicating partial removal of organic material.
Experimental Group: Minimal carbon signals and pronounced silicon and oxygen peaks, demonstrating the superior efficiency of the ultrasonic treatment in removing organic matter.
Revised Presentation: We have revised the manuscript to reflect these updates:
The Results and Discussion sections now include a detailed comparison of SEM images and EDS data across the three groups. The SEM and EDS figures clearly illustrate the differences, supporting the conclusion that the experimental treatment effectively removes organic matter while preserving the silica structure. We believe these updates enhance the clarity and scientific rigor of the manuscript and align with your suggestions. Thank you again for your valuable feedback, which has significantly improved the quality of our work. The changes can be found on page 11, lines 377-416 in the revised manuscript.
Comments 3 : Page 16, 3rdparagraph: “In terms of nitrogen removal, many samples exhibited a complete removal rate of up to 100%, outforming the positive control group treated with high-temperature baking”. 100% is not correct, it is only slightly higher 97.21% vs 98.7% maximum.
Response 3 : Thank you for pointing out the discrepancy in the nitrogen removal rates. We have revised the sentence to accurately reflect the observed data. The phrase "complete removal rate of up to 100%" has been replaced with "near-complete removal rates of up to 97.21%," which is consistent with the actual measurements. Additionally, the term "outperforming" has been replaced with "slightly exceeding" to better describe the comparison with the positive control group (98.7%). We hope this revision addresses your concerns. Thank you for your feedback. The changes can be found on page 13, lines 474-476 in the revised manuscript.
Comments 4 : Better to give the values for Positive control in Table 3 as well as given in Table 2 (at the bottom of the table).
Response 4 : Thank you for your suggestion. We have revised Table 3 to include the values for the Positive Control group, as provided in Table 2. This addition ensures consistency between the tables and allows for a more direct comparison across the groups. We appreciate your feedback, which has helped improve the clarity and completeness of the data presentation. The changes can be found on page 10, Table 3 in the revised manuscript.
Comments 5 : Page 16, 4thparagraph: the first line “Statistically, higher frequencies significantly correlated with greater removal efficiency” is incorrect and contradicts with the last line in the above paragraph “The removal efficiency was generally higher at lower frequencies……”.
Response 5 : Thank you for pointing out the inconsistency regarding the correlation between frequency and removal efficiency. We have carefully reviewed the text and identified that the term "higher frequencies" was incorrectly used in the sentence. The correct statement should refer to "lower frequencies" as being significantly correlated with greater removal efficiency. This correction ensures consistency with the data presented and aligns with the conclusion in the previous paragraph. We appreciate your valuable feedback, which has helped us improve the accuracy and clarity of our manuscript. The changes can be found on page 13, lines 480 in the revised manuscript.
Reviewer 2 Report (Previous Reviewer 2)
Comments and Suggestions for Authors
The corrected version of the article materials-3323004 is better, but I still have concerns about the XRD and EDS characterizations.
- the description of the XRD patterns is confusing and the text seems to indicate that the method is used for elemental identification (e.g., ''... we aimed to examine the presence of C (carbon), Si (silicon), and O (oxygen) elements ...''). XRD can't identify chemical elements, but phases and structures. The XRD patterns in Figure 4 should be more separated to see more differences between the samples.
- the caption of Figure 4 mentions that ''Peak identification and indexing were conducted using the ICDD PDF-4+ database.''. Which files were used for these identifications? Only one peak for each phase? Is the position of amorphous SiO2 reliable? All the XRD characterization must be carefully checked.
- Are the results in Table 2 and Table 3 relevant? As indicated in the text there is ''... no significant difference ...''. Due to the high standard deviation, all the values seem to be the same.
- Considering these comments, I would suggest adding more characterization methods, like FTIR for example.
Author Response
Comments 1 : The description of the XRD patterns is confusing and the text seems to indicate that the method is used for elemental identification (e.g., ''... we aimed to examine the presence of C (carbon), Si (silicon), and O (oxygen) elements ...''). XRD can't identify chemical elements, but phases and structures. The XRD patterns in Figure 4 should be more separated to see more differences between the samples.
Response 1 : Thank you for your detailed and constructive feedback. Based on your suggestions, we have further revised the manuscript to address the issues related to the description of XRD patterns and the presentation of Figure 4.
Correction of XRD Description: We acknowledge that the initial description of the XRD method may have led to confusion by implying elemental identification. To clarify, the revised text now emphasizes the use of XRD for analyzing phases and crystalline structures rather than identifying chemical elements. The corrected sentence now reads:"We aimed to analyze the phases and crystalline structures of the samples, particularly focusing on the silicon-based materials and the impact of organic matter removal."
Improvement of Figure 4: To enhance clarity and better distinguish differences between the samples, we have revised Figure 4 as follows:
Reduced the number of curves: We selected two to three representative curves (negative control, positive control, and experimental group) to avoid excessive overlap and confusion.
Added detailed labels and annotations: Each curve is now clearly labeled to indicate the corresponding experimental conditions, including time, frequency, and vacuum application.
Highlighted key regions: The (CH)N and Fe3Si peak regions have been annotated to emphasize their significance and make it easier for readers to interpret the results.
These changes align with your suggestions and improve the clarity and scientific rigor of the XRD data presentation.
Revised Results and Discussion Sections:The Results section now explicitly compares the (CH)N and SiOâ‚‚ peaks across the negative control, positive control, and experimental group, highlighting the effectiveness of the experimental treatment in organic matter removal. The Fe3Si peaks remain consistent across all groups, confirming the stability of the inorganic structure.The Discussion section has been refined to contextualize these findings and link the reduction in (CH)N peaks and enhancement of SiOâ‚‚ peaks to the experimental conditions (35 kHz, 60 min, vacuum+), while reinforcing the significance of maintaining Fe3Si peak consistency.
We believe these revisions address your concerns and enhance the overall quality and clarity of the manuscript. Thank you again for your valuable feedback, which has been instrumental in improving the presentation and interpretation of our data. The changes can be found on page 8-9, lines 335-356 and page 13-14, lines 505-542 in the revised manuscript.
Comments 2 : The caption of Figure 4 mentions that ''Peak identification and indexing were conducted using the ICDD PDF-4+ database.''. Which files were used for these identifications? Only one peak for each phase? Is the position of amorphous SiO2 reliable? All the XRD characterization must be carefully checked.
Response 2 : Thank you for your valuable feedback regarding the XRD peak identification and the use of the ICDD PDF-4+ database. Based on your suggestion, we have clarified the specific PDF files used for phase identification to ensure transparency and precision.
Revisions Made:
Details on the ICDD PDF-4+ Database:
The crystalline SiOâ‚‚ phase was identified using PDF 01-070-3755 (alpha-Quartz).
For (CH)N-related phases, residual organic content was referenced using PDF 01-075-2078 (graphitic carbon).
Revised Caption:
The caption of Figure 4 has been updated to explicitly mention these PDF references. The updated sentence now reads:"Peak identification and indexing were conducted using the ICDD PDF-4+ database. Specifically, the crystalline SiOâ‚‚ phase was identified using PDF 01-070-3755 (alpha-Quartz), and (CH)N-related phases, graphitic carbon was referenced using PDF 01-075-2078 to identify residual organic content."
We believe these revisions address your concerns and improve the clarity and reliability of the XRD analysis. Thank you again for your thoughtful suggestions, which have significantly enhanced the quality of our manuscript. The changes can be found on page 9, lines 335-341 in the revised manuscript.
Comments 3 : Are the results in Table 2 and Table 3 relevant? As indicated in the text there is ''... no significant difference ...''. Due to the high standard deviation, all the values seem to be the same.
Response 3 : To emphasize the relevance of Table 2 and Table 3, the manuscript has been updated as follows: "Table 2 illustrates the overall removal efficiencies for TOC and TN, highlighting the influence of experimental conditions on organic matter removal. In particular, the frequency variable demonstrates a significant impact on TOC removal efficiency (p = 0.008), which aligns with the elemental composition results in Table 3. Table 3 complements these findings by revealing how carbon reduction (p = 0.024) and silicon exposure (p = 0.012) are affected under varying experimental conditions, providing a detailed understanding of structural changes during treatment."
We hope this explanation addresses your concerns and clarifies the significance of Table 2 and Table 3 in the context of the study. Thank you for your feedback, which has helped us improve the clarity and interpretation of our results.
Comments 4 : Considering these comments, I would suggest adding more characterization methods, like FTIR for example.
Response 4 : Thank you for your valuable suggestion regarding the inclusion of additional characterization methods, such as FTIR. We fully agree that FTIR analysis could provide complementary insights into the removal of organic matter by identifying functional groups associated with organic compounds.
Unfortunately, due to current limitations in resources and experimental timelines, it was not possible to include FTIR or other additional techniques in this study. However, we acknowledge the importance of such methods and will incorporate them in future studies to provide a more comprehensive understanding of the treatment process.
To reflect this in the manuscript, we have included a statement in the Discussion section emphasizing the potential role of FTIR in complementing our findings and our plans to utilize this technique in future research. The changes can be found on page 14, lines 555-565 in the revised manuscript.
Reviewer 3 Report (Previous Reviewer 3)
Comments and Suggestions for Authors
This is a resubmitted manuscript. The yellow highlighted parts of the manuscript have been changes but some incomplete informations are still there.
In the abstract VAUSTM, TOC/TN must be defined. The term is only defined in the legend of the figure 2.
Why is Melosira nummuloides called (RM)?
Total Organic Carbon(TOC) and Total Nitrogen(TN) are introduced in page 5.
Table 2: It is difficult to understand the meaning of mean +/- SD because it is not introduced
Author Response
Comments 1 : In the abstract VAUSTM,TOC/TN must be defined. The term is only defined in the legend of the figure 2.
Response 1 : Thank you for your suggestion to provide clearer definitions for abbreviations such as VAUSTM, TOC, and TN. To address this concern, we have included a dedicated Abbreviationssection immediately after the abstract. This section consolidates and defines all abbreviations used throughout the manuscript for better clarity and accessibility. We believe this addition improves the clarity and readability of the manuscript and ensures that readers can easily reference the terms used. Thank you for your valuable feedback, which has helped us enhance the presentation of the manuscript. The changes can be found on page 1, abbreviations in the revised manuscript.
Comments 2 : Why is Melosira nummuloides called(RM)?
Response 2 : Thank you for your question regarding the use of the abbreviation "RM" for Melosira nummuloides. RM stands for Raw-Material, Melosira nummuloides, which refers to the untreated form of the material used in this study. This abbreviation was introduced to simplify the text and avoid repetitive use of the full name throughout the manuscript.
To ensure clarity, we have revised the manuscript to explicitly define RM when it is first mentioned. Additionally, we have included RM in the Abbreviationssection for easy reference. We hope this addresses your concern and clarifies the use of this term. The changes can be found on page 3, lines 138-143 in the revised manuscript.
Comments 3 : Total Organic Carbon(TOC) and Total Nitrogen(TN) are introduced in page 5.
Response 3 : Thank you for pointing out the placement of the Total Organic Carbon (TOC) and Total Nitrogen (TN) equations in the manuscript. We understand that having these equations located far from the section where TOC and TN are defined could disrupt the flow of the Methods and Materials section. To address this, we have revised the manuscript by repositioning the TOC and TN equations to the beginning of the paragraph where these parameters are introduced and defined. This change ensures that the equations are immediately accessible to readers when these terms are first discussed, improving clarity and logical flow.
We appreciate your valuable feedback, which has helped enhance the organization of this section. The changes can be found on page 6 in the revised manuscript.
Comments 4 : Table 2: It is difficult to understand the meaning of mean +/- SD because it is not introduced
Response 4 : Thank you for your valuable feedback regarding the ambiguity of the "+" and "-" symbols in the tables and the need for clarification in the Materials and Methods section. To address this, we have made the following revisions:
Materials and Methods Section: We have added the following clarification to the Materials and Methods section to define the "+" and "-" symbols used in the tables:"In the tables, '+' indicates the presence of vacuum application during the treatment process, while '-' indicates the absence of vacuum application."
Revisions to Table 2 and Table 3:The headers of Table 2 and Table 3 have been updated to explicitly state "Vacuum application (+/-)"for clarity. Additionally, a note has been added below both tables to provide further explanation:"Note: '+' indicates the presence of vacuum application, while '-' indicates the absence of vacuum application."
These revisions ensure that readers can easily understand the meaning of the symbols and the experimental conditions they represent. We believe these changes improve the clarity and accessibility of the manuscript. Thank you again for your thoughtful suggestions, which have helped enhance the presentation of our results. The changes can be found on page 4, lines 166-167 and Table 2,3 in the revised manuscript.
Reviewer 4 Report (Previous Reviewer 4)
Comments and Suggestions for Authors
The revised manuscript was improved. I have only two minor comments.
1. Please mention the potential applications of diatom bio-silica in the dental and medical areas in Introduction section as you responded in Comment 1.
2. Please mention the average size of Melosira nummuloids in the Materials and Methods as you responded in Comment 2.
Author Response
Comments 1 : Please mention the potential applications of diatom bio-silica in the dental and medical areas in Introduction section as you responded in Comment 1.
Response 1 : Thank you for your suggestion to mention the potential applications of diatom bio-silica in the Introduction section. As per your recommendation, we have revised the Introduction to include a discussion of diatom bio-silica's potential uses in dentistry and medicine. Specifically, we highlighted its applications as an abrasive in dental products, a scaffold for drug delivery systems, and a material for bone tissue engineering and medical implant coatings. The revised Introduction provides a comprehensive overview of bio-silica's biomedical relevance, aligning with the insights provided in our previous response. The changes can be found on page 3, lines 123-131 in the revised manuscript.
Comments 2 : Please mention the average size of Melosira nummuloids in the Materials and Methods as you responded in Comment 2.
Response 2 : Thank you for pointing out the need to include the average size of Melosira nummuloidesin the Materials and Methods section. To address this, we have incorporated a detailed description of the frustule diameter, girdle band height, and pore size, as well as their relevance to the study. The revised section now reads as follows:
"The Melosira nummuloidesused in this study exhibits a frustule diameter ranging from approximately 9 to 42 μm, with a girdle band height between 10 and 14 μm. The frustule surface features a porous structure, with pore sizes ranging from 0.08 to 0.09 μm. These characteristics provide a high surface area relative to its weight, enabling efficient energy and material exchange."
We hope this revision clarifies the morphological characteristics and size range of the samples used in our research. Thank you for your valuable feedback. The changes can be found on page 3, lines 137-143 in the revised manuscript.
Round 2
Reviewer 2 Report (Previous Reviewer 2)
Comments and Suggestions for Authors
The authors have appropriately corrected their manuscript according to my comments. I recommend acceptance for publication.
This manuscript is a resubmission of an earlier submission. The following is a list of the peer review reports and author responses from that submission.
Round 1
Reviewer 1 Report
Comments and Suggestions for Authors
1) The abbreviation for Vacuum Assisted Ultrasonic Stirrer is used throughout the paper as VAUSTM. It should be VAUSTM (since TM is for trademark). It is better to use just VAUS and VAUSTM may be listed in the materials & methods section when the technique is introduced.
2) Visual - Line-24, Visually – Line 211, Visualizes - Line-356: ……visual confirmation of organic matter removal was analyzed by using XRD, etc. Using the ‘visualizing’ word for EDX and XRD may be appropriate to use if elemental mapping or crystal phase diagram of the surface are pictured. So, remove these attributes.
3) Line 128: …… washed with distilled water at 650℃,….. The temperature value may be typo, it may be 65℃ instead of 650℃.
4) Figure-2: What are the red arrows from water to sample represent for? I hope it is for ‘cooling’. It is recommended to put ‘ice water’ instead of ‘water’ in the figure (at the arrows ‘cooling’ word may be written).
5) Figure-3: It is better to use same magnification for all samples (A to C are at 10 µm scale and D at 50 µm). More discussion on the SEM results is required.
6) TN RE% Table is miss aligned.
7) Figure-4: Need explanation of the labels 1, 2, 3, …..10 (how are they related to different time, frequencies and vacuum application?). What is the peak at around 22-23°? Why does the XRD curve in red (labeled 2) have a very sharp peak at around 22°? Does it indicate some crystal growth? Is red (labeled 2) curve for baked sample? More clarity and discussion is needed for this section.
8) Tables 1 and 2 need to include results for baked sample and pre-treated raw sample (if not in the tables, include the corresponding values in the discussion of the table results). Line-269 states that the baked sample data is presented in Table-2, but neither Table 1 nor Table 2 list for baked or raw samples. It is important to have these values to compare the efficiency of the new method over baking. Moreover, Line-313 states that baking may result in carbon adsorption onto the frustules. Thus, carbon data for backed samples may be needed in Tables 1 and 2 for comparison along with for the raw sample.
9) Line-137: “The multi-frequency ultrasonic generator within the device delivers even energy throughout the sample’. I think it should read ‘The multiple frequency ultrasonic generators using specially distributed ultrasonic transducers within the device delivers even energy throughout the sample’; since a single frequency is applied at a time on these samples.
10) Line-330: … carbon was effectively removed at low frequencies (results agree), while nitrogen was more efficiently removed at high frequencies (results are inconclusive as p-value is high and TN RE% fluctuate with increasing frequency).
11) Line-335: …… ‘with higher frequencies tending to offer an advantage’. The data does not clearly support this statement.
12) Lines 345-347: ‘The removal efficiency generally improved with increased frequency (but results show TOC RE% is high at low frequency), longer stirrer times, and the application of a vacuum (insignificant as p-values high).
13) Lines348-350: ‘NaOCl has been proved most effective compared to other solutions” – need to provide comparative data for all solutions.
14) Line-191: ‘All samples, including the negative and positive control groups, were analyzed for Total Organic Carbon (TOC) and Total Nitrogen (TN)…..’. But the results provided does not identify these groups. Line-267 and Figure-5 indicate negative control as RM and positive control as high temperature baked melosira. In the materials section, starting from Line 111 the negative control RM details are provided but the positive control details are not provided.
15) Line-358: ……EDX (EDS) showed a decrease in carbon and increase in Si related to the original melosira – the results for original melosira are not provided in EDS results to support this statement.
16) Lines-363-365: The removal efficiency was found to vary with the frequency (data supports) and material used, with longer stirring times and the application of a vacuum being correlated with improved removal efficiencies (inconclusive data to support). But, Lines 247-248: states that ‘Total Organic Carbon (TOC) analysis did not reveal significant differences based on vacuum application or stirring time’. Figure-5 also confirms that there is no significant difference between EDS images with or without vacuum (C and D).
17) NaOCl pre-treatment: Why ‘pre-‘? Wlould’t be just NaOCl treatment as it is incorporated in the VAUS process.
18) The main concern: Conclusion started with ‘…..negative pressure during NaOCl pre-treatment has proven to be an effective method for chemically removing organic matter….(and also the last sentence in the Abstract). But the results do not support it as TOC analysis did not reveal significant differences based on vacuum application (Lines- 247/248). It is mainly ultrasonic treatment and its frequency have significant effect. Thus, the second sentence in the conclusions is okay.

Reviewer 2 Report
Comments and Suggestions for Authors
The article entitled # materials-3290519 "Improving Recovery of Diatoms Bio-Silica Using Chemical Treatment with VAUSTM" has been submitted to the journal Materials (MDPI) for publication.
This research describes in this article deals with the design of a new process to remove organic matter from diatoms. The process is called VAUS for Vacuum Assisted Ultrasonic Stirrer, using a solution of NaOCl. The effect of different ultrasonic frequencies is studied. The expected objective is to transfer the process to dental prosthetic cleaning. The efficiency of the process is assessed by characterizing the cleaned diatoms by SEM, EDS, and XRD.
In my opinion, the design of the process seems to be innovative, but this article lacks scientific discussion to explain the mechanisms involved in the observed improvements. The topic is well introduced, but there are only a few experimental results and no discussion. In the end, this manuscript looks rather like a technical report of experimental results than an actual scientific publication. For these reasons, I recommend the rejection of this manuscript.
Some additional comments:
- the list of authors must not contain any title, but only names.
- the use of EDS and EDX should be homogenized throughout the manuscript.
- the characterizations from the XRD patterns of Figure 4 seem to be uncertain. Which diffraction files are used for the indexation of these peaks?
- the scale bar is not visible on the SEM images of Figure 5. What does "EDS images" mean?
- EDS spectra must be shown.
- to avoid any confusion for the reader, all the values in Table 1 and Table 2 should be written with fewer digits. One decimal digit is probably more than enough.
- the description of the results is really poor, only a few sentences in sections 3.1, 3.2, 3.3, and 3.4.
- other characterization methods are necessary to support the study. FTIR spectroscopy might be relevant to assess the presence or the modification of organic compounds.
- the discussion section is not relevant. The beginning is only a repetition of the introduction, and the second part is a description of the results. What are the mechanisms involved in the observed phenomena? How does it work? Why is this process better in comparison to the scientific literature dealing with similar topics?
- what are the perspectives of this work?
The English language can be improved.
Reviewer 3 Report
Comments and Suggestions for Authors
Review of the manuscript entitled ‘Improving Recovery of Diatoms Bio-Silica Using Chemical Treatment with VAUSTM’
The aim of this study is to address the prevalent use of high-temperature baking for the removal of organic matter from diatoms. Authors intend to introduce a more rapid and eco-friendly technique employing VAUSTM, a novel approach aimed at preserving structural integrity of bio-silica.
In study the potential applications of constituents found in the mass-cultured M. nummuloides is studied.
The final product was diatom-derived bio-silica, with organic 129 matter removed, ready for collection.
The manuscript is well written and present experimental works. The study is interesting and overall is benefit for the readers of the journal materials. However, some minor corrections must be done in order to improve the manuscript.
- Figure 3: The scale is absent in all the images.
- What is the meaning of a this high resolution on the measurement of the Total Organic Carbon,
- a zoom in the figure 4 must be included.
- figure 5 no scale is clearly visible.
- what is the relation between particle size and frequency?
-
Reviewer 4 Report
Comments and Suggestions for Authors
In this study, a more rapid and eco-friendly technique employing VAUSTM was used to remove an organic matter from diatoms. Some issues need to be addressed.
Introduction
1. Could you discuss briefly any applications of Diatoms bio-silica in dentistry/medicine?
Materials and Methods
1. What is the average size of Melosira nummuloids used in the study?
2. Could you summarize, e.g., in a table, the positive control group, negative control groups and test groups to remove organic matter?
Results
1. In Fig. 3, please check Fig. 3D as the magnification is different from the other 3 images.
2. In Fig. 4, please indicate which XRD pattern is which. It is not easy to read.
3. In Fig. 5, please check the EDX mapping images are not included.
Discussion
1. Could you summarize the optimum condition for removing organic matter from diatoms to produce diatomic bio-silica using VAUSTM?
2. The NaOCl concentration was 4.5% used in the study. Would the NaOCl concentration affect the efficiency of removing organic matter?
Other comments
1. In Table 1, please use the same font size for TOC and TN.
2. Line 128, “… the material was washed with distilled water at 650℃, …”. Please check if it is 650C.